# Organisational Model and Coverage of At-Home COVID-19 Vaccination in an Italian Urban Context

**DOI:** 10.3390/vaccines9111256

**Published:** 2021-10-29

**Authors:** Elettra Carini, Chiara Cadeddu, Carolina Castagna, Mario Cesare Nurchis, Teresa Eleonora Lanza, Adriano Grossi, Andrea Barbara, Svetlana Axelrod, Mauro Goletti, Paolo Parente

**Affiliations:** 1Health Directorate, Local Health Authority Asl Roma 1, 00193 Rome, Italy; elettra.carini1@gmail.com (E.C.); adriano.grossi@yahoo.it (A.G.); andrea.barbara@aslroma1.it (A.B.); mauro.goletti@aslroma1.it (M.G.); paolo.parente@aslroma1.it (P.P.); 2Section of Hygiene, Department of Life Sciences and Public Health, Università Cattolica del Sacro Cuore, 00168 Rome, Italy; carolina.castagna@gmail.com (C.C.); lanza_teresaeleonora@libero.it (T.E.L.); 3Department of Woman and Child Health and Public Health, Fondazione Policlinico Universitario A, Gemelli IRCCS, 00168 Rome, Italy; nurchismario@gmail.com; 4Institute of Leadership, University of Sechenov, 119435 Moscow, Russia; akselrod@who.int

**Keywords:** at-home vaccination, immunisation campaign, SARS-CoV-2, organisational model

## Abstract

The COVID-19 pandemic called for a reorganisation of the methods for providing health services. The aim of this paper is to describe the organisational model implemented by one of Rome’s Local Health Units (LHU), ASL Roma 1, for the “at-home COVID-19 vaccination campaign” dedicated to a target population and to outline data related to vaccination coverage stratified by health districts. A cross-sectional study was designed to describe the strategies implemented by LHU to deliver at-home vaccination programmes. People eligible for the at-home vaccination programme included patients living in the area of the LHU, being assisted by the district home care centre or not transportable or individuals with social situations that make traveling difficult. Priority for vaccination was given to (I) age > 80 years, (II) ventilated patients with no age limit, (III) very seriously disabled people with no age limit. Patients’ data were acquired from regional and LHU databases. From 5 February until the 16 May, 6127 people got at least one dose of Pfizer-Biontech Comirnaty^®^ vaccine, while 5278 (86.14%) completed the necessary two doses. The highest number of vaccines was administered during the first week of April, reaching 1296 doses overall. The number of vaccines administered were similar across the districts. The average number of people vaccinated at home was 6 per 1000 inhabitants in the LHU. This model proved to be extremely complex but effective, reaching satisfying results in terms of vaccination coverage.

## 1. Introduction

The COVID-19 pandemic has been the most challenging health emergency of this century. Among its consequences, there has been a profound reorganisation of the methods for providing health services. The academic, industry, and government sectors worked tightly together to develop and test a variety of vaccines at an unprecedented pace [1].

On 27 December 2020, the “Vaccination Day” was held in Italy, as well as throughout the European countries, as a symbolic start of the COVID-19 vaccination campaign. On 26 December, 9750 doses of vaccine arrived from Belgium at the Spallanzani Hospital hub in Rome. Then, in order to allow the entire country to participate in the European Vaccination Day, the Italian army took them for distribution to all other regions [2]. 

Afterwards, on 19 January 2021, the European Commission defined actions for strengthening the response against pandemic and accelerating the launch of vaccination campaigns, with the aim of vaccinating at least 80% of individuals over 80 years and 80% of healthcare workers and social-health workers in each Member State by March 2021 [3].

Italy’s Strategic Plan for SARS-CoV-2/COVID-19 vaccination consists of two documents: the first one titled “Elements of preparation of the vaccination strategy”, presented by the Minister of Health to Parliament on the 2 December 2020 (i.e., Decree 2 January 2021) [4]; the second named “Interim recommendations on the target groups of the anti-SARS-CoV-2/COVID-19 vaccination” dated 10 March 2021, in which the categories of the population to be vaccinated and the priorities were updated by [5].

This plan firstly defined the vaccination strategies, organisational models, staff training, logistics, characteristics of the support information system and all activities connected with vaccination, aspects relating to communication, vaccine-surveillance, and models of impact and economic analysis for the launch of the vaccination campaign [6].

Based on the European directives, the Italian COVID-19 vaccination schedule established the priority involvement of specific categories, based on professional profiles and ages. Individuals with clinical frailties were priority categories as they are among the most exposed to potential damage deriving from severe disease in case of SARS-CoV-2 infection, but also the ones with the most frequent requests for outpatient healthcare services, hospital admissions, and immunosuppressive pharmacological treatments that substantially increase the risk of contagion [7,8]. The condition of clinical frailty identifies a state of vulnerability associated with negative health outcomes, disability, and large use of health resources as a result of aging, polymorbidity, polypharmacy, and social isolation [9].

For these categories, direct access to vaccination centres might be difficult, but cannot be neglected as highly necessary to fight the pandemic.

In Italy, the vaccination campaign started by immunising the healthcare personnel and, from the beginning of February 2021, was extended to other target populations represented by, in order of access, people aged 80 and over, patients with high clinical frailty, and then people aged over 70. In order to respond to patients’ needs, our Local Health Unit (i.e., ASL Roma 1) launched on 5 February 2021 an at-home COVID-19 vaccination campaign dedicated to all non-self-sufficient people, experiencing also innovative methods in order to achieve faster and more efficient organisational models [10,11]. The Metropolitan city of Rome is organised in six LHUs (i.e., ASL Roma 1–6) and is, in turn, split into six health districts (i.e., I, II, III, XIII, XIV, XV). The total area covered is 525.6 km2 (i.e., 40.8% of the overall city) and the total population served is 1,026,479 people (i.e., 36% of the population of Rome) [12].

Globally, evidence regarding at-home vaccination programmes are now scarcely available, neither for COVID-19 nor for other vaccine-preventable diseases. A study conducted by Bond et al. in 1998 [13] aimed to ascertain the effectiveness of an at-home vaccination service for children behind in their vaccination schedule in Australia. Furthermore, in India, at-home vaccination services, dedicated to child immunisation, are easily accessible and have been available for years [14]. In addition, in India, the issue of the introduction of at-home COVID-19 vaccination programmes directed to frail elderly individuals who were bedridden or with severe motor difficulties has been much debated [15].

This paper is aimed at describing the organisational strategies implemented by the Local Health Unit for the promotion and the delivery of an at-home COVID-19 vaccination campaign for the target population and at outlining the related differences in terms of administered doses.

Furthermore, we reported data related to vaccination coverage of patients, who benefited from the at-home vaccination service, stratified by health districts.

## 2. Materials and Methods

A cross-sectional study was designed to provide a descriptive analysis of the strategies implemented by LHU to deliver at-home vaccination programmes, to highlight strengths and weaknesses of such an organisational model, and to measure vaccination coverage in the target population.

The three methods of vaccination delivery are hereafter briefly described:

ASL Roma 1 teams—a doctor and a nurse of the health districts staff.Special continuity of care units—teams of professionals that were established at regional level for the pandemic to assist people who tested positive for COVID-19 at home. As their primary necessity decreased over time, they were assigned to the vaccination campaign as well.Home care providers—teams working for the LHU with external contracts, mainly to assure home care.

Moreover, ASL Roma 1 teams and special continuity of care were also provided with UBER service that drove the professionals to the patients’ houses. The use of UBER allowed to overcome the double problem of the lack of corporate cars and of the lack of parking spots in the city that slowed the process of reaching the houses of the people in need in the necessary timeframe for the stability of the vaccine (i.e., 6 h in dedicated coolers).

### 2.1. Population Definition and Setting 

People eligible for the at-home vaccination programme included patients being assisted by the district Home Care Service, not transportable (e.g., bedridden, with reduced mobility) or individuals with social situations that make traveling difficult (e.g., absence of caregivers, architectural barriers). Among these two different clusters, priority for vaccination was given, in the following order, to these categories: (I) age > 80 years, (II) ventilated patients with no age limit, (III) very seriously disabled people with no age limit [16], (IV) extremely vulnerable patients [5] and, to follow, all the (V) other subsequent age groups.

In order to benefit from the above-mentioned programme, the request could have been made by the patient/caregiver calling the regional toll-free numbers or sending a direct request (e.g., email, call, and request on site) to the health district. In addition, the general practitioners were actively invited to notify the people in need.

Prerequisites for the at-home service was living in the territory of the LHU, even without residency, but having a general practitioner in the Lazio region was a necessary condition.

### 2.2. Data Sources

Patients’ data were acquired from regional and LHU databases. The information available for all the patients was name and surname, date of birth, fiscal code, telephone number, address, general practitioner. All the data concerning the number of doses administered and the modality of administration were collected from the starting date of the at-home vaccination programme (i.e., 5 February 2021) until 16 May 2021. The data collected after this date are not included in this study since, on 17 May 2021, the national recommendations on the administration of second doses changed (i.e., the term for the second doses was shifted from 21 to 35 days). 

### 2.3. Statistical Analysis

A descriptive analysis was conducted, adopting graphs and tables, to provide basic summaries about the number of doses administered and the number of people vaccinated per health district and provider. In addition, a quantitative analysis of the three methods of vaccination delivery was conducted. Given the non-normal distribution of the considered variables (i.e., number of doses and type of provider), two nonparametric tests were run. The Kruskal–Wallis test was adopted to investigate whether there was a statistically significant difference between the medians of the three providers in terms of administered doses, while the Mann–Whitney U test was performed to individually compare the three providers, investigating if a specific provider led to a difference in administered doses. All analyses were conducted using the software STATA 16 (StataCorp LP, College Station, TX, USA). 

## 3. Results

The only vaccine used for the at-home program was Pfizer-Biontech Comirnaty^®^. From 5th February until the 16 May, 6127 people got at least one dose of vaccine, while 5278 (86.14%) completed the necessary two doses (i.e., with at least 21 days latency between the two doses, as indicated in the institutional recommendations) [17]. The highest number of vaccines was administered during the first week of April (i.e., week 14), reaching 1296 doses overall (Figure 1).

Comparing the health districts, the number of vaccines administered (in terms of first and second doses) are similar across the districts. The health district III administered the highest number of vaccines 2109 (18.5% of the total 11,405 –first plus second- doses administered), followed by health district XIV (2039 vaccinations; 17.9%), health district XIII (2032 vaccinations; 17.8%), health district II (1965 vaccines; 17.2%), health district I (1776; 15.6%), and health district XV (1484; 13.0%). Most of the health districts had their peak of vaccination during the 14th, 15th, and 16th weeks, as shown in Figure 2.

Examining more in detail the provision modalities, special continuity of care units performed 6195 administrations of vaccines overall (i.e., first and second doses), while ASL Roma 1 teams registered 2822 administrations and home care providers 2388 doses. Figure 3 shows in columns the number of doses administered per health district and detailed according to the three provision modalities. 

Special continuity of care units was the method of administration of vaccines most commonly used, administering 1270 vaccines for health district III and 1247 doses for health district II. ASL Roma 1 teams were the most common method of administering vaccines only for health district XIII (869 doses); all the other health districts had more doses given by special continuity of care units. Home care providers were more similarly employed by all the districts and administered from 310 to 462 doses per health district. Comparing the at-home intervention to the population of the entire health district, the average number of people vaccinated at home was 6.0 per 1000 inhabitants in the LHU, ranging from a minimum 4.9 per 1000 registered in the health district XV to a maximum 8.2 of the health district XIII (Table 1).

The Kruskal–Wallis test revealed that there was a statistically significant difference (X^2^ = 237.42; *p* = 0.0001) in median administered doses across the three providers. Results from the Mann–Whitney U test showed that the median numbers of administered doses were statistically different between ASL Roma 1 teams and home care providers (z = −14.84; *p* < 0.01), implying a significant impact of ASL Roma 1 teams on administered doses. The same test highlighted that the median numbers of administered doses were statistically different also between home care providers and special continuity of care units (z = 16.78; *p* < 0.01), showing a significant impact of special continuity of care units on administered doses. In conclusion, based on the findings from the third Mann–Whitney U test, the median number of administered doses between ASL Roma 1 teams and special continuity of care units was not statistically different at any level smaller than 22.6%.

## 4. Discussion

At the end of 2020, during the second wave of the COVID-19 pandemic, in Italy the health districts were involved in the vaccination of the general population as soon as the central government gave the possibility to start the vaccination campaign. Bearing in mind some of the duties of the health district (e.g., active engagement of patients to reduce inequities and the at-home assistance), as soon as the immunisation programme was opened to the population aged 80 and over, the LHU planned an at-home vaccination campaign to reach all the patients unable to go to the vaccination hubs. Therefore, the at-home vaccination campaign started with a launching day on 5 February 2021 in the health districts III and XIV and by the next week all the others joined them. For the first two weeks, the only personnel involved in the vaccination were the medical doctors and nurses of the health district, but as soon as the magnitude of the effort was clear, special continuity of care units and home care providers were also involved in this programme.

Thanks to the extensive work undertaken by all the professionals involved and the management from central coordination, by 16 May among the 6127 people vaccinated, 86.14% had completed the two doses. Overall, the health district III performed the highest number of doses (2109; 18.5%) and the peak of administration was reached during weeks 14, 15, 16. Generally, the health districts had many vaccinations done by the special continuity of care units. The at-home vaccination program reached 6.0 per 1000 inhabitants of the LHU. Among the three providers, a significant difference in administered doses was found (X^2^ = 237.42; *p* = 0.0001). 

The nature of employment contracts and working time may explain the differences, in terms of administered doses, across the three providers. Particularly, the ASL Roma 1 team was also in charge of the back office work: contacting all the people to give them the appointment with one of the providers, registration of the vaccination in the corporate and regional databases, and vial requests. All of this affected the number of possible vaccinations that the ASL Roma 1 team could perform, thus the need for extra teams. Special continuity of care and home care providers were asked to perform as many vaccinations as possible and, according to their possibilities, all the needed vials of vaccine were provided. Special continuity of care teams were paid by the hour, while home care providers by performance.

Mass vaccination campaigns are not a new argument in healthcare. It relates to a rapid vaccination intervention across age groups which requires careful planning to be successful. Other than planning, the key elements are represented by vaccine supply, communication, and surveillance [18]. Indeed, the most important aspect that made the at-home vaccination campaign successful was the planning.

Israel represents the gold standard when addressing the COVID-19 immunisation program. The main facilitating factors described—apart for the intrinsic national characteristics—include organisational and logistical capacities of community-based healthcare providers; centralised national system; well-trained, salaried, community-based nurses; cooperation between government, health plans, hospitals, and emergency care providers; support tools and decision-making frameworks. Other factors are related to the pandemic and include the mobilisation of special government funding, simple and easily implementable criteria for determining who had the priority, and well-tailored efforts to encourage people to sign up and then show up to get vaccinated [19].

The vaccination campaign followed different organisational models also in Israel with the opening of vaccine-designed areas, mobile units for remote places, and also for people confined at home. The coordination of immunisation campaigns by the central government, and specific incentives (i.e., the release of a ‘green pass’) have been an important booster to the vaccination uptake [20].

The joint reading of the current encouraging findings allows for different main implications. This program of at-home vaccination has a great social value. In fact, in the absence of such an organisational model, fragile categories would at best have received their COVID-19 vaccination with great delay. Alternatively, receiving vaccination in hospital would have exposed this fragile population to serious risks of contagion, as well as difficulties in transportation. Not to mention that, in the worst case, they could have chosen not to receive the vaccination at all, with all the serious consequences that this would have brought.

Given that implementing an at-home COVID-19 vaccination program on a short time frame is a complex undertaking, the organisational model used to reach all the people in need, at the time scheduled by the regional age-related timeline, required the cooperation of different services and professionals. Indeed, a multidisciplinary working group was a key element to overcoming the challenges and successfully concluding the campaign. The experts agreed that, when considering the use of vaccines in emergencies, a multidisciplinary approach is essential and that the prevention and control of the infectious diseases should be envisaged within the larger context of public-health priorities in times of crisis [21].

The health planning process, needed to deliver such a complex service directly to the patients’ home, represents a public health tool, albeit preliminary, useful to pave the way and steer choices of decision-makers in the establishment of priorities related to vaccination campaigns in emergency situations. 

To the best of our knowledge, this research represents not only the first attempt to illustrate a model of an at-home vaccination campaign, but also this model itself was the first one implemented in Italy and probably, to date, the only documented one in the world. However, this work should be considered in the light of its limitations. 

Firstly, neither stratification by age groups nor by main diseases was performed, making it not possible to analyse the population with other variables, especially health characteristics. Secondly, as far as there is not a census of the people needing at-home care, it was not possible to fully verify if all the people in need received/accepted the service. Lastly, the data available and, on which our considerations are based on, are those only belonging to the LHU. Moreover, the organisational model described should be considered closely connected to the current Italian socio-health context and therefore it is possible that it is not comparable to the health contexts of different countries worldwide.

Further studies are then needed in order to consider both health outcomes and costs related to at-home vaccination campaigns. Furthermore, additional economic evaluations are required to better investigate the sustainability and cost-effectiveness of alternative ways to deliver at-home vaccination campaigns.

## 5. Conclusions

This cross-sectional study was designed to illustrate the strategies implemented by ASL Roma 1 in order to perform the first Italian at-home vaccination programme, highlighting strengths and weaknesses of such an organisational model. This model proved to be extremely complex but similarly effective in its characteristics and allowed health districts to reach all those frail patients who otherwise would not have been able to travel to vaccination hubs. This immunisation campaign achieved a great result reaching 6.0 per 1000 inhabitants of the ASL Roma 1, which represents a remarkable outcome in terms of vaccination coverage.

## Figures and Tables

**Figure 1 vaccines-09-01256-f001:**
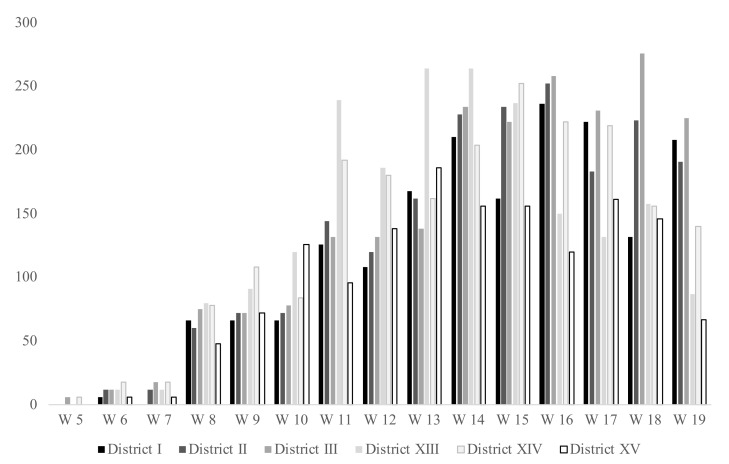
Number of doses (first and second doses) administered per week by the LHU.

**Figure 2 vaccines-09-01256-f002:**
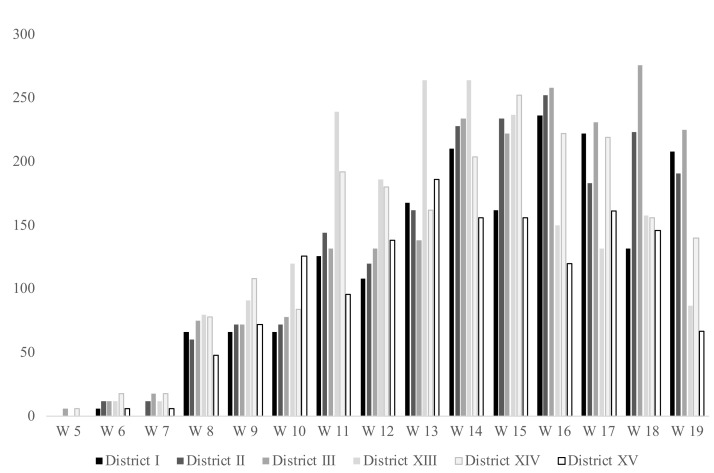
Number of doses (first and second doses) administered per week and per health district.

**Figure 3 vaccines-09-01256-f003:**
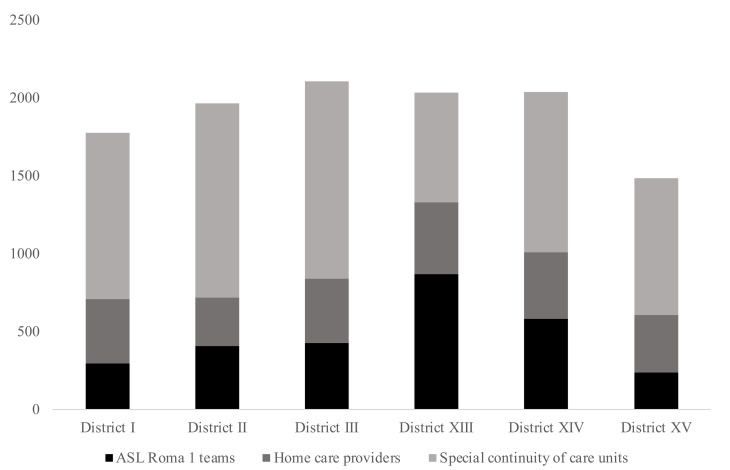
Number of doses (first and second doses) administered per health district and per delivery mode.

**Table 1 vaccines-09-01256-t001:** Number of people (only first dose) vaccinated per health district/ASL Roma 1 and per provider. People vaccinated per 1000 inhabitants.

	ASL Roma 1 Teams	Home Care Providers	Special Continuity of Care Units	Total	Population	Vaccinated People/1000 Inhabitants
District I	150	228	605	983	167,330	5.9
District II	206	168	657	1031	167,649	6.1
District III	222	246	699	1167	205,759	5.7
District XIII	490	239	360	1089	133,388	8.2
District XIV	288	222	564	1074	191,851	5.6
District XV	123	195	465	783	160,502	4.9
ASL Roma 1	1479	1298	3350	6127	1,026,479	6.0

## Data Availability

The data that support the findings of this study are available on request from the corresponding author, C.C. (Chiara Cadeddu).

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
