# Peer review of "Organisational Model and Coverage of At-Home COVID-19 Vaccination in an Italian Urban Context"

_vaccines, 2021, doi:10.3390/vaccines9111256_

Round 1
Reviewer 1 Report
Elettra et al., described the at-home COVID-19 vaccination strategy in Roma, Italy in the beginning phase of vaccination, targeting the people who had problems going out for vaccination including patients and elder people. The process of vaccination including background, plan and methodology) and result (number of vaccinated people per districts, per periods) were exactly and carefully described.
However, the manuscript is simply the description of the process and seems not more than the public report. It was not clear if the data contributes to the vaccination strategy for current COVID-19 or future emerging infectious diseases.
Major comment
The purpose for comparisons between districts (Fig.2 ); and between ASL Roma1, Special continuity unit and home care providers (Fig. 3) were obscure.
Minor comments
Line 24. The readers are not familiar with the word ‘ASL Roma 1’. It is recommended to use other word, such as local health unit.
Line 163. The total number of target person in the area should be described. i.e. ‘… 16th May, X% of the people (6127 / total number of target) got at least one dose….’
Line 173-176. What do % indicate?
Fig. 2. In comparison between the districts, please show the vaccination completion ratio (number of doses / number of total subjects involved in the program), rather than number of doses.
Reviewer 2 Report
The manuscript by Carini and colleagues outlines the protocol for an "at-home COVID-19 vaccination program (ASL Roma 1) for individuals that are less likely to get vaccinated (age > 80 year; ventilated patients with no age limit, and very seriously disabled people with no age limit. Such programs are desperately needed and hence the study provides a protocol that other municipalities could use for at-risk patients. I found that the manuscript was well-written and methodology straightforward.
Author Response
Thank you for pointing out the constructive reviews to improve our manuscript. A point-by-point response has been shown here. We have provided a detailed response to each point raised describing what amendments have been made to the manuscript text and where these can be viewed. All changes to the manuscript are indicated in the text by using the “Track Changes function”.
We agree with all comments raised and have made every effort to revise our paper in light of the comments. Please find the points and our responses below:
- The manuscript by Carini and colleagues outlines the protocol for an "at-home COVID-19 vaccination program (ASL Roma 1) for individuals that are less likely to get vaccinated (age > 80 year; ventilated patients with no age limit, and very seriously disabled people with no age limit. Such programs are desperately needed and hence the study provides a protocol that other municipalities could use for at-risk patients. I found that the manuscript was well-written and methodology straightforward.
We really thank the reviewer for this comment. We are glad that the manuscript was appreciated.
Reviewer 3 Report
The authors describe a complex organizational model utilized by ASL Roma for at-home vaccination campaign in Italy. The study revealed the success of the at home vaccination campaign for the at risk population. The manuscript is well written, easy to follow with the limitations clearly stated.
Author Response
Thank you for pointing out the constructive reviews to improve our manuscript. A point-by-point response has been shown here. We have provided a detailed response to each point raised describing what amendments have been made to the manuscript text and where these can be viewed. All changes to the manuscript are indicated in the text by using the “Track Changes function”.
We agree with all comments raised and have made every effort to revise our paper in light of the comments. Please find the points and our responses below:
- The authors describe a complex organizational model utilized by ASL Roma for at-home vaccination campaign in Italy. The study revealed the success of the at home vaccination campaign for the at risk population. The manuscript is well written, easy to follow with the limitations clearly stated.
We really thank the reviewer for this comment. We are glad that the manuscript was appreciated.
Round 2
Reviewer 1 Report
The revised manuscript by Elettra et al. is significantly improved.
The current manuscript should be recommended for publication.